# The Effect of Online Supervised Exercise throughout Pregnancy on the Prevention of Gestational Diabetes in Healthy Pregnant Women during COVID-19 Pandemic: A Randomized Clinical Trial

**DOI:** 10.3390/ijerph192114104

**Published:** 2022-10-28

**Authors:** Ane Uria-Minguito, Cristina Silva-José, Miguel Sánchez-Polán, Ángeles Díaz-Blanco, Fátima García-Benasach, Vanessa Carrero Martínez, Irune Alzola, Ruben Barakat

**Affiliations:** 1AFIPE Research Group, Faculty of Physical Activity and Sport Sciences-INEF, Universidad Politécnica de Madrid, 28040 Madrid, Spain; 2Gynecology and Obstetrics Department, Hospital Universitario Severo Ochoa de Leganés, 28911 Leganes, Spain; 3Gynecology and Obstetrics Department, Hospital Universitario Puerta de Hierro de Majadahonda, 28222 Majadahonda, Spain; 4Clíninca Zuatzu, 20018 San Sebastian, Spain

**Keywords:** pregnancy, gestational diabetes mellitus, weight gain, physical exercise

## Abstract

(1) Background: to examine the effect of an online supervised exercise program during pregnancy on the prevention of GDM, and on maternal and childbirth outcomes. (2) Methods: we conducted a randomized clinical trial (NCT04563065) in 260 pregnant women without obstetric contraindications who were randomized into two study groups: intervention group (IG, N = 130) or control group (CG, N = 130). An online supervised exercise program was conducted from 8–10 to 38–39 weeks of pregnancy. (3) Results: no significant differences were found at baseline in maternal characteristics; nevertheless, certain outcomes showed a favorable trend towards the IG. A lower number and percentage of GDM cases were found in the IG compared to the CG (N = 5/4.9% vs. N = 17/16.8%, *p* = 0.006). Similarly, fewer cases of excessive maternal weight gain (N = 12/11.8% vs. N = 31/30.7%, *p* = 0.001) were found in the IG, and a lower percentage of instrumental deliveries (N = 8/11.3% vs. N = 13/15.1%) and c-sections (N = 7/9.9% vs. N = 20/23.3%, *p* = 0.046). (4) Conclusions: an online supervised exercise program can be a preventative tool for GDM in healthy pregnant women.

## 1. Introduction

During healthy pregnancies, in order to guarantee adequate fetal growth and development, insulin sensitivity is reduced [1]. As a result, the capacity for glucose uptake is compromised and blood glucose levels increase [2]. Normally, this reduced insulin sensitivity and hyperglycemia are counteracted by a greater insulin secretion causing a state of hyperinsulinemia. However, if pancreatic beta-cells are incapable of producing sufficient insulin [1], gestational diabetes mellitus (GDM), one of the most common complications of pregnancy [3], will develop.

GDM is commonly defined as a carbohydrate intolerance diagnosed for the first time in pregnancy [1], between the second and third trimester, and which generally resolves after delivery [4,5]. The worldwide prevalence of GDM is estimated to be of 14.2% [6] and a recent meta-analysis estimated that around 12.3% of South European women have GDM [7].

GDM can lead to serious short- and long-term consequences in the mother, fetus, and newborn. Women who develop GDM are at greater risk of suffering cardiovascular diseases, preeclampsia, preterm birth, and caesarean delivery, in addition to birth canal lacerations [8]. However, the most important maternal side effect of GDM is the continuation of diabetes after delivery [9]. The chance of progressing to type 2 diabetes mellitus (T2DM) after GDM increases by 70% within 28 years postpartum [10]. Similarly, there are increased risks of macrosomia at birth (birth weight ≥ 4 kg) [4], labor difficulties such as birth trauma, fracture, and shoulder dystocia, and neonatal hypoglycemia [8] in pregnancies under GDM. Furthermore, research shows that offspring of GDM mothers are at a greater risk of developing T2DM [11] and obesity [12] in the future, in addition to insulin resistance [11] and cardiovascular diseases [8]. In some cases, where GDM is untreated, perinatal mortality may also occur [8].

Various risk factors have been associated with the development of GDM. These include being over 35 years old, previous GDM or delivery of macrosomic baby, polycystic ovarian syndrome, use of corticosteroids, or belonging to a high-risk population (Aboriginal, Hispanic, Asian, or African) [13]. Nevertheless, the major risk factors for developing GDM are being overweight or obese (body mass index (BMI) ≥ 25 kg/m^2^) at the beginning of pregnancy [13,14], and excessive gestational weight gain [15].

According to the scientific literature, to reduce the risk of excessive maternal weight gain and thus, the risk of suffering GDM, regular physical activity (PA) is strongly recommended, in the absence of any obstetric contraindications [16]. In addition, it has been suggested that PA can reduce the risk of GDM by 31% [17]. However, despite the multiple health benefits demonstrated by the practice of physical activity [16], it has been observed that pregnant women reduce their activity levels with pregnancy [18]. International PA guidelines [19,20,21] recommend the equivalent of 150 min/week of moderate-intensity physical activity during pregnancy and, specifically, a recent meta-analysis suggested an accumulation of >90 min/week of moderate-intensity PA to reduce the odds of GDM by 46% [22].

The COVID-19 global pandemic outbreak and the lockdown measures imposed by governments in March 2020 have contributed to a more sedentary lifestyle and to a significant reduction in physical activity levels among pregnant women [23], which may lead to numerous chronic pathologies such as cardiovascular diseases, type 2 diabetes [24], and metabolic syndrome, which includes obesity and dyslipidaemia [25], and pregnancy disorders including preeclampsia [26]. Consequently, an increase in the prevalence of GDM was observed among pregnant women in 2020 compared to pregnant women in 2019 when there were no mobility restrictions [27].

Given these data and the dangerous health consequences associated with GDM for both mother and fetus, an adequate preventive strategy is absolutely necessary.

Although the scientific literature is not entirely conclusive, many pre-pandemic experimental studies show the positive influence of supervised exercise on prevention of GDM in healthy pregnancy [28,29].

Therefore, during the ongoing pandemic and with the technological progress of online modalities, the aim of this study was to examine the influence of a supervised online exercise program throughout pregnancy on gestational diabetes. We hypothesized that a supervised, moderate regular exercise program throughout pregnancy using an online model may help prevent the development of gestational diabetes.

The aim of this study was to examine the effect of an online supervised exercise program during pregnancy on the prevention of GDM, and on maternal and childbirth outcomes.

## 2. Materials and Methods

### 2.1. Study Design

The present RCT (Identifier: NCT04563065) was carried out between August 2020 and February 2022 following the ethical guidelines of the Declaration of Helsinki, last modified in 2000.

The protocol for this study was approved by the Ethical Commission of Research of the Universidad Politécnica de Madrid and Ethical Commission of Clinical Research (CEIC) of Hospital Universitario Severo Ochoa in Leganés, Hospital Universitario Puerta de Hierro in Majadahonda, and Clinica Zuatzu in San Sebastián.

### 2.2. Participants

A total of 445 healthy pregnant women from three hospitals were recruited during their first prenatal visit at the hospital (Figure 1). They were informed about the nature of the study and were assessed for eligibility by their obstetric health providers.

Pregnant women with singleton, uncomplicated pregnancies (no type 1 or 2 diabetes at baseline), with no risk of preterm delivery and with no serious medical contraindications that prevented them from exercising, were invited to participate in the study. Women who were simultaneously participating in another trial or exercise program, planning to give birth in a different hospital, or had no medical surveillance throughout pregnancy were not included in the study. All participants signed an informed consent.

After assessing women according to the aforementioned inclusion and exclusion criteria, they were randomly allocated to either intervention group (IG) or control group (CG) employing REDCap software. A computer-generated list of random numbers was used in order to ensure a blinded randomization sequence, and the access to the REDCap software for randomization was performed by one health professional from each hospital.

### 2.3. Intervention Group

Women assigned to the intervention group received standard obstetric attention and a structured and supervised online moderate exercise intervention program three days a week (50–60 min/session) from the 8–10th week of pregnancy to the end of the third trimester, at the 38–39th week. Of the three sessions, two were carried out via Zoom 120 and the last session was carried out following YouTube videos, which had been previously 121 filmed by the physical activity professionals in our research group. The content of 122 the videos on YouTube were adapted to the weeks of pregnancy.

The exercise sessions were supervised by a qualified physical activity and sport science professional. The exercise program consisted of complementary and interrelated activities following a methodological model divided into seven components adapted to each trimester of pregnancy, previously established by our research group [30]:physical and emotional activation (gradual warm-up);aerobic exercises;light muscle strengthening;coordination and balance;stretching and relaxation exercises;pelvic floor muscle strengthening;final talk.

During the exercise sessions women used a heart rate (HR) monitor. HR was kept under 65–70% of the age-predicted maximum and rate of perceived exertion scale ranged from 12 to 14 (somewhat hard).

### 2.4. Control Group

Women in the control group attended regular scheduled visits to their obstetricians and midwives, usually every 4–5 weeks until the 36–38th week of gestation and then weekly until delivery. Despite being in the control group, women were not discouraged from exercising during pregnancy, but they were questioned every trimester about their exercise habits using a “Decision Algorithm” (by telephone) [31] to control their PA levels.

This algorithm contained three questions about the follow-up, frequency, and volume of physical activity pregnant women carried out weekly. If pregnant women exercised excessively, that is, they followed a supervised or autonomous physical activity program three or more days a week, with a duration of over 20 min a day, they were kept out of the study.

Additionally, these women received physical activity and nutritional recommendations throughout pregnancy. They were asked about the amount of exercise they practiced once every trimester.

### 2.5. Outcomes

Data collected from the pregnant mother before pregnancy, during, and after pregnancy, in addition to data collected from the fetus and newborn, were obtained from the SELENE platform, which is used to manage all hospital obstetric records.

### 2.6. Primary Outcome

The primary outcome was the diagnosis of GDM. The National Diabetes Data Group criteria were followed and included a 50 g maternal glucose screen (MGS) at 24–28 weeks of gestation, which was used to determine plasma glucose 1 h after a 50 g load of glucose administered orally. The screen test was considered positive when the values were equal or greater than 140 mg/dL. If the screen test was positive, the women were required to have a fasted oral glucose tolerance test (OGTT) before 30 weeks of gestation with data taken from medical records: 100 g of glucose load with blood samples taken fasted and at 1, 2 and 3 h post-glucose ingestion. Positive results were fasting glucose equal to or greater than 95 mg/dL, or values equal to or greater than 180 mg/dL at 1 h, equal to or greater than 155 mg/dL at 2 h, or equal to or greater than 140 mg/dL at 3h. Diagnosis of GDM occurred if there were at least two abnormal results in the OGTT.

### 2.7. Secondary Outcomes

Maternal characteristics collected were maternal age, weight, height, pre-pregnancy BMI, parity, smoking during pregnancy, occupation, and previous miscarriage. BMI was calculated as weight (kg) divided by height (m^2^), and individuals were classified as underweight (BMI < 18.5 kg/m^2^), normal weight (BMI < 18.5 to 24.9 kg/m^2^), overweight (BMI ≥ 25 to 29.9 kg/m^2^), and obese (BMI ≥ 30 kg/m^2^) [32].

Gestational weight gain was calculated based on pre-pregnancy weight and weight on the last medical visit before delivery. Childbirth data included maternal gestational age, mode of delivery (non-instrumental, instrumental, or c-section), birth weight, birth length, head circumference, and Apgar Scores 1 and 5 min. Excessive maternal weight gain was defined according to pre-pregnancy BMI [33]. If pre-pregnancy BMI was below 18.5 kg/m^2^, the healthy weight gain range was between 12.5 and 18 kg; if BMI was between 18.5 and 24.9 kg/m^2^, the recommendation was between 11.5 and 16 kg; if BMI was between 25 and 29.9 kg/m^2^, the healthy weight gain recommendation was between 7 and 11.5 kg; and if BMI was greater than 30 kg/m^2^, the weight gain should be between 5 and 9 kg [33]. All data were collected at the first prenatal visit by reviewing the medical records.

### 2.8. Power Sample Calculation

For the primary outcome (gestational diabetes mellitus) [34], power calculations were based on earlier studies, and a prevalence of 8% and 25% was used in the intervention and usual care group, respectively. Following a two-sample comparison (χ^2^) with a 5% level of significance and a statistical power of 0.90, the study population was set to 98 participants in each group. Assuming 25% lost to follow-up, we recruited 130 participants for each study group.

### 2.9. Statistical Analysis

Version 25.0 of IBM SPSS for Windows (IBM Corporation, Armonk, NY, USA) was utilized for the statistical analysis. Screening for the hypothesis of normality was performed using the Kolmogorov–Smirnov test.

Pearson’s chi-square test was used to compare maternal characteristics at baseline (number of participants by BMI subcategory, parity, maternal smoking during pregnancy, occupation, and previous miscarriages), cases of diagnosed gestational diabetes mellitus, number of abnormal glucose screenings, number of excessive maternal weight gains according to pre-pregnancy BMI, and mode of delivery between groups. Standardized adjusted residuals were used to complete and interpret with precision the observed association. Binary logistic regressions were used to assess risk estimation (odds ratio (OR)) of GDM in relation to IG and CG.

Independent t-tests were conducted to assess differences in maternal age, weight, height, maternal weight and gestational weight gain at glucose screening, final maternal weight, final gestational weight gain, gestational age at the beginning of the program, at glucose screening, and at delivery, basal glucose at 50 g OGTT, O’Sullivan values, basal glucose at 75 g OGTT, 60 min OGTT, 120 in OGTT, 180 min OGTT, birth weight, birth length, head circumference, Apgar Scores 1 and 5 min, between the IG and CG. This same test was used within GDM diagnosis or not for maternal weight at glucose screening, final maternal weight birth weight, birth length, head circumference, and Apgar Scores 1 and 5 min between IG and CG. The effect size was obtained using Cohen’s d index.

Data for continuous variables are shown as means and standard deviations, and categorical variables are shown as frequencies and percentages. Analyses were performed under intention-to-treat principles. The risk level was set at 0.05.

## 3. Results

### 3.1. Study Population

In total, 445 women were assessed for eligibility, out of which 185 were excluded: 123 did not satisfy the inclusion criteria, 17 declined to participate, 6 suffered prior diabetes mellitus, and 39 claimed other reasons. Participants were randomized into IG (n = 130) or CG (n = 130). Twenty-eight pregnant women were lost to follow-up within the IG, 11 had low adherence, 8 changed hospitals, 6 had missing glucose screening, and 3 alleged other reasons. Within the CG, 29 pregnant women were lost to follow-up: 3 had persistent bleeding, 9 changed hospitals, 8 did not comply with the decision algorithm, 4 had missing glucose screening, and 5 alleged other reasons. As a result, 102 individuals in the IG and 101 in the CG were analyzed (Figure 1).

### 3.2. Baseline Maternal Characteristics

Table 1 shows the baseline characteristics of the pregnant women. No significant differences (*p* > 0.05) were found between groups in maternal characteristics. However, BMI and smoking habits data showed a favorable trend towards the IG acting as a potential predictor of the main outcome. The gestational age at the beginning of the program was 10.24 ± 2.94 weeks.

### 3.3. GDM Cases

There were significant differences between the IG and CG (χ^2^_(6)_ = 7.474; *p* = 0.006; Vc = 0.188) in gestational diabetes (Table 2). In an intent-to-treat analysis, the prevalence of GDM was 4.9% (5/102) in the intervention group and 16.8% (17/101) in the control group (odds ratio [OR], 0.255; 95% confidence interval [CI] [0.090, 0.720]; *p* = 0.010). This represents a 74.5% reduction in the prevalence of GDM.

### 3.4. Comparison of 50 g OGTT and 75 g OGTT Glucose Values between IG and CG

Significant differences were found in the number of abnormal glucose screenings (χ^2^_(3)_ = 3.969; *p* = 0.046; Vc = 0.140) between IG and CG. Twenty-four pregnant women (23.8%) in the IG and 37 (36.6%) in the control group underwent the 75 g OGTT. IG had lower blood glucose levels at basal, 60, 120, and 180 min in the postintervention 75 g OGTT compared to the control group (*p* < 0.05) (Table 3).

### 3.5. Maternal Outcomes

There were significant differences in final maternal weight between IG and CG (t_(186)_ = −2.504; *p* = 0.0.013; d = −0.402) and in excessive weight gain (χ^2^_(9)_ = 10.891; *p* = 0.001; Vc = 0.232) (Table 4).

### 3.6. Childbirth Outcomes

There were significant differences in mode of delivery between the IG and CG (χ^2^_(6)_ = 6.155; *p* = 0.046; Vc = 0.198), with c-sections (23.3% vs. 9.9%) and instrumental deliveries (13.1% vs. 11.3%) occurring more often in the CG (Table 5). No other significant differences were found in childbirth outcomes.

### 3.7. Maternal and Childbirth Outcomes in Women without GDM

After dividing the sample according to the diagnosis of GDM, significant differences in maternal final weight were found in the group without GDM (t_(149)_ = 2.333; *p* = 0.020; d = −0.364) (Table 6).

### 3.8. Maternal and Childbirth Outcomes in Women Diagnosed with GDM

Table 7 shows the results for women who had been diagnosed with GDM. Significant differences were found in weight at 28 weeks (t_(12)_ = −2.222; *p* = 0.046; d = −1.056), weight at the end of pregnancy (t_(16)_ = −2.292; *p* = 0.035; d = −0.807), birth length (t_(10)_ = −2.615; *p* = 0.026; d = −2.131) and head circumference (t_(10)_ = −3.115; *p* = 0.038; d = −1.594).

## 4. Discussion

The purpose of the present study was to investigate the effect of an online structured exercise program throughout pregnancy on the prevention of GDM. This innovative method consisted of online group exercise sessions which combined different forms of physical exercise (aerobic, strength conditioning, coordination, or flexibility). Our findings suggest that the pregnant population can significantly benefit from joining an online group training session throughout pregnancy during the COVID-19 pandemic.

A major finding of this study is the significantly lower diagnosis of GDM in the IG compared to CG (4.9% vs. 16.8%, respectively) and the reduction in its overall risk by 74.5%. These results are in accordance with previous studies conducted by Koivusalo et al., where the prevalence of GDM in the IG was significantly lower compared to the CG (13.9% vs. 21.6%, respectively, *p* = 0.044) [35]. Similarly, Barakat et al. in 2018 showed that the prevalence of GDM was significantly higher in the CG compared to the IG (6.8% vs. 2.6%, respectively, *p* = 0.033) [28]. Possible mechanisms by which physical exercise reduces the risk of GDM include the increased energy expenditure and glucose consumption during exercise. During exercise, due to muscle contraction, a greater translocation of glucose transporter-4 (GLUT-4) to the muscle cell’s surface occurs. As a result, glucose uptake from the bloodstream increases, thereby reducing blood glucose levels and insulin resistance by not requiring as much insulin [36]. Similarly, exercise increases adiponectin concentration, which improves cellular sensitivity to insulin, and reduces inflammatory markers associated with insulin resistance [37]. Additionally, it has been demonstrated that physical exercise during pregnancy is effective in reducing maternal weight gain, which is one of the main risk factors in GDM [28].

This study also found significant differences in the mode of delivery between groups (*p* = 0.046), wherein instrumental and caesarean deliveries were more common among women in the CG than in IG (13.1% vs. 11.3%; 23.3% vs. 9.9%, respectively). These results are similar to previous literature where the percentage of caesarean and instrumental delivery was lower in the exercise group compared to control [38]. In addition, Sanda et al. found that the odds of caesarean delivery were significantly lower in the high active group compared to the low active group, and that being in the high active group was associated with higher odds of vaginal delivery [39]. The results obtained in our study could be explained by the COVID-19 [23] pandemic, which may have further increased the sedentary behavior among pregnant women in the CG.

Further, final maternal weight and excessive maternal weight gain were higher in the CG. Previous interventions, and a recent systematic review and meta-analysis, also conclude that exercise interventions are effective in preventing excessive maternal weight gain [28,40,41]. Reduced leptin concentrations in active women may be the mechanism behind reduced weight gain. Nevertheless, these results should be interpreted with caution because there is a limited number of RCTs which have studied the effect of physical exercise on maternal weight gain.

Moreover, our results showed higher values in the 50 g MGS in the CG, which called for the need to carry out the OGTT, with the associated higher health costs and the greater discomfort of women exposed to this 3 h test.

Furthermore, overall glucose levels were higher in the CG and more specifically, these were significantly higher in the 75 g OGTT measurements (basal, 60, 120, and 180 min). Our intervention has shown that exercise can be effective for maintaining lower glucose levels, which has also been demonstrated in previous literature

In addition, of the women diagnosed with GDM, pregnant women in the CG had higher weight at the time of measurement of diabetes and at the end of pregnancy, and their newborn had a larger birth length and head circumference. Thus, these data suggest that exercise during pregnancy may serve as a tool for controlling final weight and excessive weight gain in gestational diabetic mothers and their offspring.

Our findings indicate further benefits of exercise for the healthy pregnant population, which may be essential to reduce the prevalence of GDM and to prevent future comorbidities such as type 2 diabetes, cardiovascular diseases, or preeclampsia in the mother, in addition to birth trauma, hypoglycemia, or macrosomia in the newborn, and type 2 diabetes, obesity, or cardiovascular diseases in the offspring [4,7,8,9,11,12].

Overall, our findings confirm the results of a previous systematic review and meta-analysis, which concluded that exercise during pregnancy has a significant protective effect against GDM [16]. However, further data on the relationship between maternal exercise, GDM, and the COVID-19 situation are needed.

We believe that this is the first study to connect an online structured and supervised exercise program with high adherence to the reduction in the prevalence and diagnosis of GDM, and the control and management of GDM limiting future comorbidities and ensuring an optimal quality of life throughout pregnancy, postpartum, and even the lifetime. Moreover, this RCT, which was intended as an intervention focused on lifestyle, confirms the benefit of physical exercise as a determinant element for the prevention and management of gestational diabetes mellitus. In summary, performing a physical exercise intervention with comprehensive lifestyle patterns may be necessary for the health of the pregnant population during a pandemic.

In conclusion, an online supervised exercise program throughout pregnancy, aiming to address the COVID-19 limitation, reduced rates and prevalence of GDM in healthy pregnant women. These results should be used with caution to recommend supervised exercise during healthy pregnancy as a preventive tool for GDM.

## 5. Conclusions

An online supervised exercise program carried out throughout pregnancy was found 347 to benefit pregnant women by reducing the risk of developing common pregnancy diseases 348 such as GDM.

## 6. Limitations

A possible limitation of our study was the lack of nutritional control and evaluation. Nonetheless, healthy lifestyle counseling including nutritional advise was provided during obstetric care visits throughout pregnancy. In addition, the supervision of the correct performance of exercises is a limitation compared to face-to-face activities; however, the technological resources used allowed an adaptation forced by the COVID-19 pandemic, and this may be the trend in future studies. Another important limitation was the baseline data of some variables (BMI and smoking habits), which could be predictors of the final outcomes.

## Figures and Tables

**Figure 1 ijerph-19-14104-f001:**
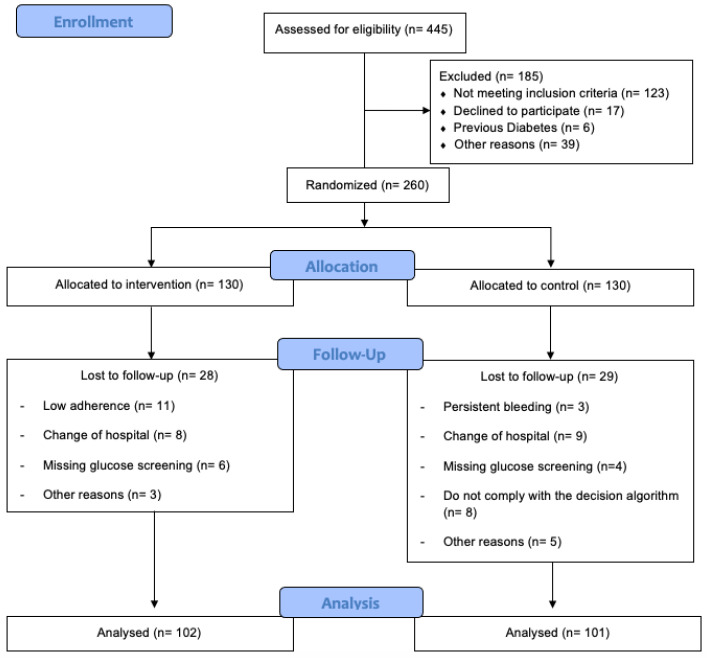
Study population flow chart.

**Table 1 ijerph-19-14104-t001:** Maternal characteristics at baseline.

Maternal Characteristics			
	IG (n = 102)	CG (n = 101)	*p*-Value
Age (years)	33.80 ± 3.27	33.29 ± 5.27	0.422
Maternal Height (m)	1.64 ± 0.06	1.62 ± 0.06	0.100
Maternal weight (kg)	61.80 ±10.37	64.90 ± 14.07	0.084
BMI (n/%)	22.70 ± 4.17	25.09 ± 5.40	
<18.5	5/5.0	3/3.0	
18.5 to 24.9	72/72.0	61/60.4	0.133
25 to 29.9	16/16.0	21/28.8	
>30	7/7.0	16/15.8	
Parity (n/%)			
None	70/65.6	60/59.4	
One	26/25.5	33/32.7	0.391
Two or more	6/5.9	8/7.9	
Smoking during pregnancy			
No	96/95.0	89/88.1	0.076
Yes	5/5.0	12/11.9	
Occupation (n/%)			
Active job	50/49.0	45/44.6	
Sedentary job	35/34.3	26/25.7	0.075
Homemaker	17/16.7	30/29.7	
Previous miscarriage (n/%)			
None	72/70.6	64/63.4	
One	26/25.5	26/25.7	0.155
Two or more	4/3.9	11/10.9	

Data are expressed as mean ± SD, unless otherwise indicated.

**Table 2 ijerph-19-14104-t002:** Cases of gestational diabetes in intervention and control groups.

	IG (N = 102)	CG (N = 101)	*p*-Value
Gestational Diabetes (n/%)			
No	97/95.1	84/83.2	0.006
*Standardized adjusted residuals*	2.7	−2.7	
Yes	5/4.9	17/16.8	
*Standardized adjusted residuals*	−2.7	2.7	

**Table 3 ijerph-19-14104-t003:** Glucose values for the 50g OGTT and 75 g OGTT between IG and CG.

	IG (n = 102)	CG (n = 101)	*p*-Value
Gestational age at screening	27.94 ± 1.76	28.32 ± 1.07	0.178
Basal 50 g MGS (mg·dL^−1^) *	79 ± 6.41	80.99 ± 6.96	0.136
50-g MGS or screening glucose value or O’Sullivan	126 ± 23.98	127.44 ± 27.79	0.724
Basal 75 OGTT (mg·dL^−1^)	78.24 ± 7.45	84.96 ± 11.73	0.033
60 min OGTT **	133.72 ± 29.42	157.33 ± 29.74	0.018
120 min OGTT	124.61 ± 23.25	142.10 ± 27.55	0.041
180 min OGTT	98.16 ± 43.80	127.52 ± 43.46	0.040

* Milligrams per deciliter (mg/dL). ** OGTT: oral glucose tolerance test.

**Table 4 ijerph-19-14104-t004:** Maternal outcomes.

	IG (n = 102)	CG (n = 101)	*p*-Value
Maternal weight at glucose test	70.57 ± 12.57	74 ± 12.71	0.132
Gestational weight gain (kg) at glucose test	7.60 ± 13.65	8 ± 12.99	0.852
Final maternal weight (kg)	72.68 ± 10.72	77.35 ± 12.48	0.013
Final gestational weight gain (kg)	11.46 ± 8.57	11.18 ± 5.15	0.787
Excessive weight gain (n/%)			
No	90/88.2	70/69.3	
*Standarized adjusted residuals*	3.3	−3.3	0.001
Yes	12/11.8	31/30.7	
*Standarized adjusted residuals*	−3.3	3.3	

**Table 5 ijerph-19-14104-t005:** Childbirth outcomes.

	IG (n = 102)	CG (n = 101)	*p*-Value
Gestational age at delivery	39.54 ± 3.22	39.12 ± 1.51	0.288
Birth weight (g)	3263.16 ± 434.73	3202.98 ± 495.80	0.438
Birth length (cm)	49.96 ± 1.87	49.77 ± 2.29	0.194
Head circumference (cm)	34.64 ± 1.42	34.38 ± 1.29	0.268
Apgar 1	8.79 ± 0.60	8.93 ± 0.74	0.140
Apgar 5	9.90 ± 0.34	10 ± 1.04	0.097
Mode of delivery (n/%)			
Non-instrumental	58/78.9	53/61.6	
* Standarized adjusted residuals*	2.3	−2.3	
Instrumental	8/11.3	13/15.1	
* Standarized adjusted residuals*	−0.7	0.7	
C-section	7/9.9	20/23.3	0.046
* Standarized adjusted residuals*	−2.2	2.2	

**Table 6 ijerph-19-14104-t006:** Women without GDM.

	IG (n = 97)	CG (n = 84)	*p*-Value
Maternal weight at screening	71.15 ± 12.67	73.87 ± 14.44	0.249
Final weight (kg)	73.19 ± 10.73	77.78 ± 14.44	0.020
Birth weight (g)	3266.09 ± 446.57	3206.57 ± 485.67	0.464
Birth length (cm)	50.01 ± 1.89	48.57 ± 2.43	0.318
Head circumference (cm)	34.70 ± 1.42	34.39 ± 1.37	0.272
Apgar 1	8.78 ± 0.61	8.99 ± 0.63	0.061
Apgar 5	9.90 ± 0.35	9.04 ± 0.96	0.307

**Table 7 ijerph-19-14104-t007:** Women diagnosed with GDM.

	IG (n = 5)	CG (n = 17)	*p*-Value
Weight at week 28	61.25 ± 5.74	74.95 ± 17.26	0.046
Final weight (kg)	63.8 ± 5.96	74.84 ± 15.01	0.035
Birth weight (g)	3216.25 ± 172.40	3180.45 ± 580.82	0.107
Birth length (cm)	48.75 ± 0.35	50.75 ± 1.03	0.026
Head circumference (cm)	33.25 ± 0.35	34.30 ± 0.72	0.038
Apgar 1	9 ± 0.00	8.62 ± 1.29	0.595
Apgar 5	10 ± 0.00	9.77 ± 0.60	0.527

## Data Availability

Not available.

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
