# Peer review of "The Effect of Online Supervised Exercise throughout Pregnancy on the Prevention of Gestational Diabetes in Healthy Pregnant Women during COVID-19 Pandemic: A Randomized Clinical Trial"

_ijerph, 2022, doi:10.3390/ijerph192114104_

Round 1

Reviewer 1 Report

The article concerns a very current topic, i.e. the increase in GDM incidence in connection with the COVID-19 pandemic. The time of the pandemic has increased the use of various online classes / courses. This form of classes seems to be particularly useful for pregnant women. The form of intervention described in the article should be more widely promoted - the obtained results support such a recommendation. From the point of view of the intervention, a slightly weaker point is the limited control of the reliability of exercises performed by pregnant women. A minor technical note: the last two sentences of the text seem to be a quote from a review - to be checked and developed by the authors.

Author Response

Dear Sir or Madam,

Thank you very much for your comments. We will make the necessary changes in order to improve the points you mentioned.

Kind regards

Reviewer 2 Report

This is a well-written paper about a well-designed study that is very timely. 

I have some very minor suggestions and one important concern.   line 83 pregnancy 84 and with the 88 may help prevent 141 why were excess exercisers kept out; how was excess defined? 159-160 Is this the clinical definition of GDM?   only important concern Many, many individual tests of significance were performed, increasing the probability of a type I error. Might it be necessary to adjust the alpha value for significance to take these into account? Residuals aren't telling us much in a 2x2 table It might be instructive to use a logistic regression to model the probability of GDM from (some) of the many potential explanatory variables in Tables 1, 3 and 4. Similarly, it might be informative to model birth outcomes from the explanatory variables in Tables 6 and 7.   line 191 screenings lines 210-218 repeats information in Fig. 1 236 omit In this sense 237 omit Besides 269 that the pregnant population can benefit from joining an online group training session 283 not helps, reduces 294 also be explained 295 omit experienced 297 not Besides, Further 305-308 may be omitted 316 not On the other hand, In addition 321 the healthy 335 not mandatory, important? omit 343-346

Author Response

Thank you very much for your comments.

We will make the changes suggested for vocabulary and omit the unnecessary lines.

However, please see the attachment for the other suggestions made.

Regards

Reviewer 3 Report

This is a much needed study, of particular interest to the scientific community and of the public particularly at time such as these.

There are however, a number of issues that have a negative effect to its potential importance.

Firstly, groups are not well matched at all. There maybe not statistical significance between them at baseline, but if you add all of the categories that this is nearly reached (BMI, smoking status etc) and which are of significance, when assessing diabetic risk, then the outcome isn't really a surprise. It is a bit sad that the research team went to all this effort during the pandemic to do this study, made proper sample size calculations (which however need a bit more detail to the text so please revise and expand as well) only to fail to do a proper randomisation, which would stratify groups according to at least a couple of them. It is also a bit of a surprise that these baseline differences, which are bound to have an effect on the outcome are not properly noted anywhere on the text. In fact these should be mentioned everywhere, including the abstract, and form the main chunk of the limitations section, which currently is non existent.

I also do have an objection on the use of stats. I am not a statistician but I don't think that binary regression is the right method to use. Considering the complexity and the co founders of the primary outcome, I wod think that GEE would probably need to be used.

Apart from these issues (which are very important nonetheless) and the need to re-read the text to avoid some bad English ("greater sedentary lifestyle" is not the right expression to use), everything else seems to be in order.

Author Response

Thank you for your comments.

Regards.

Round 2

Reviewer 3 Report

Please see my original comments as none of them has been responded in a scientific manner.

Author Response

Kind regards.
